# Study of Dynamic Accumulation in β-D-Glucan in Oat (*Avena sativa* L.) during Plant Development

**DOI:** 10.3390/polym14132668

**Published:** 2022-06-30

**Authors:** Peter Hozlár, Veronika Gregusová, Peter Nemeček, Svetlana Šliková, Michaela Havrlentová

**Affiliations:** 1National Agricultural and Food Center, Research Institute of Plant Production, Bratislavská Cesta 122, 92168 Piešťany, Slovakia; peter.hozlar@nppc.sk (P.H.); svetlana.slikova@nppc.sk (S.Š.); 2Department of Biotechnology, Faculty of Natural Sciences, University of Ss. Cyril and Methodius in Trnava, Námestie Jozefa Herdu 2, 91701 Trnava, Slovakia; gregusova4@ucm.sk; 3Department of Chemistry, Faculty of Natural Sciences, University of Ss. Cyril and Methodius in Trnava, Námestie Jozefa Herdu 2, 91701 Trnava, Slovakia; peter.nemecek@ucm.sk

**Keywords:** β-D-glucan, oat, accumulation, plant development, leaves, root

## Abstract

Oat is an important natural source of β-D-glucan. This polysaccharide of the cell wall of selected cereals is known for a number of health-promoting effects, such as reducing the level of cholesterol in the blood serum, stabilizing the level of blood glucose, or enhancing immunity. β-D-glucan has positive effects in the plant itself. There is a lack of information available, but the storage capacity of the polysaccharide and its importance as a protective substance in the plant during mild forms of biotic and abiotic stress are described. The accumulation of β-D-glucan during the ontogenetic development of oats (*Avena sativa* L.) was determined in the present work. Two naked (Valentin, Vaclav) and two hulled (Hronec, Tatran) oat varieties were used. Samples of each plant (root, stem, leaf, panicle) were collected in four stages of the plant’s development (BBCH 13, 30, 55, 71). The average content of the biopolymer was 0.29 ± 0.14% in roots, 0.32 ± 0.11% in stems, 0.48 ± 0.13% in leaves and 1.28 ± 0.79% in panicles, respectively. For root and panicle, in both hulled and naked oat varieties, sampling date was the factor of variability in the content of β-D-glucan. In stems in hulled varieties and leaves in naked varieties, neither the sampling date nor variety influenced the polysaccharide content. The content of β-D-glucan in the leaves of hulled and naked varieties decreased during the first three stages of plant development, but in the stage of milk ripeness the amount increased. The decreasing trend during milk ripeness, was also observed in the roots of both hulled and naked oats. However, in the panicle of hulled and naked oat varieties, the content of β-D-glucan increased during plant growth. Due to practical applications of natural resources of β-D-glucan and isolated β-D-glucan is useful to know the factors influencing its content as well as to ascertain the behavior of the polysaccharide during plant development.

## 1. Introduction

Oat (*Avena sativa* L.) belongs to the *Poaceae* family and, similar to the other members of the *Avena* genus, is a low-input cereal crop used worldwide for human food diet and animal feed [1]. In recent years, there has been an increased use of oat grains in the human diets due to their richness in (1,3-1,4)-β-D-glucan (hereafter as β-D-glucan), proteins, vitamins such as vitamin E, minerals (Fe, Ca) and some antioxidants such as avenanthramides [2].

Oat β-D-glucan is commonly accepted and used as a functional ingredient due to its positive effect on human health. The polysaccharide belongs to dietary fibers—high-molecular-weight carbohydrates of plant origin, which have a beneficial effect on important functions of the gastrointestinal tract and on systemic processes in the human organism [3]. As β-D-glucan is a polysaccharide resistant to digestion and absorption in the small intestine, it also attenuates blood cholesterol and glucose levels in the serum and helps to reduce the risk of cardiovascular diseases and hyperglycemic syndrome, to improve the liver functions and to reduce the extreme body weight [4,5,6]. Dietary fiber regulates the rate of nutrient digestion and their absorption [7], serves as a substrate for the microbiota of the gut [8] and promotes laxation [9]. It is also believed that insoluble oat fiber reduces the number of carcinogens in the gastrointestinal tract [10]. The U.S. Food and Drug Administration recommends a daily intake of at least 3 g of β-D-glucan from oats or barley and this recommendation is also adopted by the European Food Safety Association.

The unique functionalities of oat β-D-glucan are related to its innate property of generating highly viscous solutions in the proximal gut and thereby controlling nutrient absorption in the human body [11]. This property depends directly on the molecular weight and concentration of the β-D-glucan molecule in the solution [12]. The structure of this molecule consists of unbranched and unsubstituted chains of (1,3)- and (1,4)-β-glucosyl residues [13], so it is a polymer with β-D-glucopyranosyl units joined by β-(1→4) and β-(1→3) linkages in various ratios [14]. The structural characteristics of β-D-glucan, especially the ratio of β-(1→3) to β-(1→4) units, known as the DP3:DP4 ratio, are important determinants of viscosity and solubility [13,15]. The fine structure of the polysaccharide is responsible for its health benefits in humans [16].

β-D-glucan is most abundant in the cell walls of cereals, specifically in the starchy endosperm of the grain, where it can contribute up to 70% by weight of the walls in barley (*Hordeum vulgare* L.), rye (*Secale cereale* L.) and oats (*Avena sativa* L.) [13,17]. The content and molecular weight of oat β-D-glucan are determined by the genotype and the physiological state of the plant as well as by environmental conditions during the plant growth [11].

The accumulation of β-D-glucan is observed in the walls of endosperm cells of the developing grain and in the surrounding maternal tissues [18,19], although it has also been found in vegetative organs of the plant, namely, in the root, coleoptile, stem and leaf [20,21]. The rate of β-D-glucan accumulation was found to be higher in the initial phase of grain development and reached a maximum of about 25 days after the development of flower organs [22].

The family of genes responsible for the synthesis of β-D-glucan is called cellulose synthase such as genes (*Csl*), of which subgroups *CslF*, *CslH* and *CslJ* are responsible for the synthesis of the polysaccharide [13,23,24,25]. Several studies suggest that *CslF* genes are crucial for the synthesis of the molecule [23,24,26]. The *CslF6* is the most transcribed *CslF* gene in most barley tissues [26,27,28] and it is also associated with the production of β-D-glucan in wheat, oats and rice [13,23,25]. The expression levels of the *CslF* family have been extensively studied in barley tissues and during barley seed development [29], with members of the gene family showing significant differences in expression between different tissues. *HvCslF6* is expressed in almost all tissues and during seed development, but other members of the family appear to be more specialized, being expressed for a short time length during specific stages of grain development or in specific tissues [29]. The present study investigates the accumulation of β-D-glucan in oat (*Avena sativa* L.) different vegetative tissues during plant development.

## 2. Materials and Methods

### 2.1. Material

Two naked (Hronec and Tatran, *Avena sativa* var. *nuda* L.) and two hulled (Valentin and Vaclav, *Avena sativa* L.) varieties were used in our study. Valentin is a very early yellow-grained variety registered in the year 2008 in the Slovak Republic. It has low to medium plant height and proportion of husks, short panicles, good resistance to powdery mildew, and medium to good oat rust, grass rust and brown spot resistance. Vaclav is a yellow-grained medium-early variety registered in the Slovak Republic in 2013. Its height is short to medium and panicles are short. The variety has very good resistance to powdery mildew, good resistance to grass rust, brown spot, and oat rust. Compared to hulled oat varieties of the Slovak provenience it has a high content of β-D-glucan in the grain. Hronec is of Slovak origin with the year 2012 of registration. It is a medium-early variety with medium height and very short to short panicles. Hronec has very good resistance to powdery mildew, good resistance to grass rust and brown spot and medium to good resistance to oat rust. Tartan was bred in the Slovak Republic in 2010. It is a medium-early variety with high height and medium panicles. Its resistance to powdery mildew, oat rust, grass rust and brown spot is good.

Plants were grown in the year 2014 on the experimental fields of the National Agricultural and Food Centre, the Research Institute of Plant Production in Piešťany (Slovak Republic, 48°59′ N, 17°81′ E). Seeds were planted in March 2014, and the method of randomized blocks in five replications was used. Under field conditions, four selected plant organs (roots, leaves, stems, and immature inflorescences) were collected during four stages of plant cultivation (BBCH 13, 30, 55, 71). Representative plants in sufficient quantity to obtain enough dry samples were chosen from the middle part of the field. Plants were cleaned free from soil and dried at 45 °C for 24 h. Then, plants were divided into the following individual parts: roots, leaves (in all developmental stages), stems and panicles (only in the last stages, when they were developed). Dried samples were cut into small pieces and stored in sealable containers in the refrigerator. Right before analyses samples were milled using an ultracentrifugal mill (ZM 100, Retsch GmbH—Co.KG, Haan, Germany) to achieve 5 mm particles. The content of β-D-glucan in samples was determined in the following BBCH-scale phases: (1) Phase I: Stage 13—3 leaves unfolded, sampling 4 April 2014; (2) Phase II: Stage 30—Beginning of stem elongation: pseudostem and tillers erect, first internode begins to elongate, top of inflorescence at least 1 cm above tillering node (7 May 2014); (3) Phase III: Stage 51—Beginning of heading: tip of inflorescence emerged from sheath, first spikelet just visible (3 and 11 June for hulled and naked oats, respectively); (4) Phase IV: Stage 71—Watery ripe: first grains have reached half their final size (June 24).

### 2.2. Determination of β-D-Glucan Content

Total content of β-D-glucan in analyzed samples was determined using the Mixed-linkage β-glucan assay kit (Megazyme, Bray, Ireland) based on the method published by [30]. Approximately 10 g of the dry sample was milled to pass a 0.5 mm screen using an ultracentrifugal mill. A 100 mg-aliquot of each sample was wetted with 0.2 mL of 50% ethanol in a tube. Then, 4 mL of 20 mM sodium phosphate buffer, pH 6.5, were added and the sample was vortexed. The tube was immediately placed in a boiling water bath and incubated for 60 s. The mixture was vortexed, incubated at 100 °C for 2 min and stirred again. The tube was incubated at 50 °C for 5 min to equilibrate the temperature. Lichenase (0.2 mL, 10 U) was added to the tube and then the reaction mixture was incubated for 1 h at 50 °C with regular vigorous stirring (4 times every 15 min) on a vortex mixer. Sodium acetate buffer (5.0 mL, 200 mM, pH 4.0) was added and the content was vigorously mixed on a vortex mixer. After 5 min of equilibration at room temperature and centrifuging (1000× *g*, 10 min, 24 °C) 0.1 mL aliquots were dispensed into three 12 mL-test tubes. β-glucosidase (0.1 mL, 0.2 U) in 50 mM sodium acetate buffer, pH 4.0, was added to two of these tubes (reaction), whereas the enzyme was replaced by 0.1 mL of 50 mM sodium acetate buffer, pH 4.0, in the third tube. All tubes were incubated at 50 °C for 10 min. The GOPOD Reagent (3.0 mL) was added to each tube and incubated at 50 °C for a further 20 min. The tubes were removed from the water bath and the mixture absorbances were measured within 1 h at 510 nm against the reaction blank. The absorbance at 510 nm against reagent blank within 1 h was measured. The absorbance readings were converted into β-D-glucan content by using the following formula: $\beta \text{-}\mathrm{D-glucan}\;\left(\%\right)=\Delta A\cdot \frac{F}{W}\cdot 8.46$. F is determined by the ratio of glucose mass (100 μg) and glucose absorbance (100 μg), W is the weight of the sample expressed in mg, ∆A is the extinction after β-glucosidase treatment (reaction) minus reaction blank absorbance, and 8.46 is the conversion factor. Each sample’s β-D-glucan content was calculated as the mean of three replicates and was expressed as a percentage of the dry matter.

### 2.3. Statistical Evaluation

Obtained results were analyzed using the statistical program SPSS for Windows Release 11.5.1. One-way analysis of variance (ANOVA) was used with *post hoc* test. The least squares (LSD) method was used to evaluate the statistical significance of the differences in means (*p* ≤ 0.05).

## 3. Results

The β-D-glucan content in different ontogenetic phases during oat cultivation was determined in different plant organs (roots, stems, leaves and panicles). The oat varieties Vaclav and Valentin were representatives of the hulled oats, whereas Tatran and Hronec represented the naked ones.

### 3.1. Variability of β-D-Glucan in Varieties of Oat along Different Ontogenetic Phases

In roots, the highest content of β-D-glucan was observed in Phase I (Figure 1). Among varieties, the highest content was in Valentin (0.65%) and the lowest in Hronec (0.43%). The average content of the monitored polysaccharide was in the variety Vaclav at 0.50% and in Tatran at 0.45%. A decrease in β-D-glucan was observed in all samples during plant development compared to Phase I. In Phase II, the average content of the monitored polysaccharide was as follows: Tatran—0.27%, Valentin—0.27%, Hronec—0.22% and Vaclav—0.17%. During Phase III and Phase IV, a relatively stable content of β-D-glucan was observed in all samples, and the average content was between 0.14% and 0.28%. Statistically significant changes were noted between Phase I and IV (*p* > 0.05).

The average content of β-D-glucan in the stem was in Phase I as follows: 0.34% for Hronec—0.34%, Tatran—0.89%, Valentin—0.42%, and Vaclav—0.25% (Figure 2). In hulled oats, the β-D-glucan content increased in Phase II and in Valentin it was 0.52% and in Vaclav it was 0.43%. Naked oats showed a decrease in Phase II of 0.21% (Hronec) and 0.30% (Tatran). A decrease was observed in Tatran (0.22%), Valentin (0.29%), and Vaclav (0.19%) in Phase III, whereby an increase was detected in Hronec (0.29%). Phase IV separated the hulled and naked varieties, while in hulled oats a decrease in the content of β-D-glucan was detected compared to Phase III (Valentin—0.18%, Vaclav—0.21%) and in naked oats an increase was monitored (Hronec—0.42%, Tatran—0.37%).

The highest content of β-D-glucan in young leaves was observed in Phase I in Valentin (0.80%). The content of the polysaccharide was 0.60% in Tatran and 0.51% in both Hronec and Vaclav (Figure 3). A significant decrease was observed compared to Phase I in both hulled oats (Valentin 0.35%, Vaclav 0.36%) in Phase II as well as in naked Tatran (0.45%). In Hronec, the content of β-D-glucan increased slightly (0.52%). A decreasing trend was observed in Phase III in Tatran (0.38%), Valentin (0.29%) and Hronec (0.38%). Vaclav showed a slight increase in β-D-glucan in this phase (0.39%). A general increase in β-D-glucan was observed in Phase IV compared to Phase III in all samples (Hronec—0.60%, Tatran—0.69%, Valentin—0.38%, Vaclav—0.48%).

Accumulation of β-D-glucan was determined in oat panicles, although the panicle was not developed in Phase I in any oat varieties (Figure 4). From Phase II, an increase was observed in all samples in all analysed terms. In Phase II, hulled oats showed a higher β-D-glucan content (Valentin—0.61%, Vaclav—0.77%) compared to naked oats (Hronec—0.31%, Tatran—0.37%). The increasing trend was detected in Phase III, where the average content of β-D-glucan was 1.37% and 1.48% for naked oats Hronec and Tatran and 0.61% and 0.98% for hulled Valentin and Vaclav, respectively. The highest β-D-glucan content was recorded in all samples in Phase IV (Hronec—1.96%, Tatran—2.49, Valentin—1.96% and Vaclav—2.5%).

### 3.2. Comparison of Naked and Hulled Oat Varieties in the Accumulation of β-D-Glucan in Different Plant Tissues during Ontogenesis

The content of β-D-glucan in the root was highest in Phase I and in hulled oats (Figure 5). The average content of the polysaccharide was in Phase I in hulled oats at 0.57% and in naked oats at 0.44%. The content of β-D-glucan in the root decreased considerably during plant development and, in Phase II, naked oats contained 0.25% of β-D-glucan and hulled oats 0.24%. In Phase III, the average content of β-D-glucan was in naked oats without a change compared to Phase II (0.24%) and 0.20% in hulled oats. In Phase IV, we showed a decrease in naked oats (0.17%) and an increase in β-D-glucan content in hulled oats (0.18%).

The average β-D-glucan content in the stem during plant ontogenesis depends on the type of oat variety is shown in Figure 6. In Phase I, a higher content of monitored polysaccharide was observed in naked oats (0.43%) compared to hulled (0.34%). Differences in the content of monitored cell wall polysaccharides became apparent in Phase II. The β-D-glucan content was higher in hulled oats (0.47%) compared to naked (0.26%). In Phase III, the content of β-D-glucan decreased only slightly in naked oats (0.25%) and strongly in hulled (0.23%) compared to Phase II, and so the content of the cell wall polysaccharides was almost similar. The content of β-D-glucan increased in Phase IV in naked oats (0.39%) and decreased in hulled oats (0.19%) compared to the previous monitored phase.

Compared to the average content of β-D-glucan in the leaves of naked and hulled oats during plant development, the higher content in Phase I was detected in hulled oats (0.64%), whereas naked oats possessed 0.55% (Figure 7). In Phase II, naked oats start to accumulate higher amounts of β-D-glucan in their leaves (0.48%) compared to hulled (0.35%). In naked oats, a decrease was observed in the content of β-D-glucan (0.38%) compared to Phase II. The β-D-glucan content did not change significantly in this phase in hulled oats (0.34%). In Phase IV, an increase was detected in both groups of oats compared to the previous two phases, whereas it was much stronger in naked oats. The average content of β-D-glucan was 0.65% in naked oats and 0.43% in hulled oats.

In Phase I panicles were not formed, therefore their β-D-glucan content was analysed beginning from Phase II (Figure 8). A higher average β-D-glucan content was recorded in Phase II in hulled oats (0.69%) compared to naked (0.34%). A strong increase in the content of the monitored polysaccharide was detected in Phase III in naked oats. Panicles accumulated, on average, 1.42% of β-D-glucan. In hulled oats, also an increase in the accumulation of β-D-glucan was shown, but it was smaller as compared to naked ones (0.80%). In Phase IV, an increasing trend in β-D-glucan content was observed in both naked and hulled oats, whereas the average content was almost similar; 2.22% in naked and 2.23% in hulled oats.

According to statistical evaluation, it can be assessed that generally no statistically significant differences were observed in the content of β-D-glucan between naked and hulled oat samples of all analysed plant organs. Statistically significant differences were observed in the content of β-D-glucan in panicles during plant development and the statistical evaluation separated the final Phase IV from the other previous one. Moreover, the content of monitored polysaccharides was significantly different in panicles compared to other organs.

## 4. Discussion

The plant cell wall consists of a mechanical network of cellulosic microfibrils in a strengthened matrix of gel phase (non-cellulosic and pectic) polysaccharides [31]. It is essential for plant growth and development because it contributes to determining the functional specialization of cells by regulating their shape, permeability, and mechanical properties. It is important to understand the chemical composition and biology of cell walls because the polysaccharides that make up their cell walls, including β-D-glucan [28], also have important roles in agro-food production, human health [32] as well as in biofuel production [33].

β-D-glucan is a component of the cell wall, especially found in the endosperm of oats and barley, with a content ranging from 4 to 10% and 5 to 12% kernel dry weight, respectively [34,35], being generally lower in oats than in barley [11]. Besides grains, β-D-glucan can also be found in much lower amounts in vegetative tissues such as coleoptiles, leaves and roots [13,21,36]. Negligible distribution of β-D-glucan in the epidermal cell walls was observed in sorghum leaves [16]. Light microscopy has confirmed the widespread occurrence of the cell wall polysaccharide in maize leaves as well [20]. The accumulation of the polysaccharide at high levels was detected in the sclerenchyma fibers of developing leaf primordia in rice [21] and similar observations were also made for mature rice leaves [36] and young barley leaves [13].

In our research, β-D-glucan was measured in roots, stems, leaves and developing panicles of naked and hulled oat varieties. The amount of the polysaccharide in oat vegetative structures varied with the type of tissue. Generally, the lowest amounts were detected in the root and stem. In the root, the content of β-D-glucan was in the range from 0.17% to 0.58%, with an average of 0.29%. In the stem, the content of β-D-glucan ranged from 0.20% to 0.48% with an average of 0.32%. The content of β-D-glucan in leaves was higher and it was in the range from 0.34% to 0.65% with an average of 0.48%. The highest content of β-D-glucan was detected in developing panicles, where the average was 0.97%. β-D-glucan can be found in the cell walls of young cells. The polysaccharide in the walls of growing tissues may contribute to the elasticity of young developing cells, which is required for cell wall elongation [13,37,38].

The content of β-D-glucan in oat vegetative tissues varied in our experiment according to the developmental stage of the plant. Both, a decrease as well as an increase in the content of β-D-glucan were observed in analyzed plant organs during plant development. In roots, a decrease was detected during plant development and, generally, the content decreased from 0.51% to 0.18%. No clear trend in the accumulation of β-D-glucan was detected in stem and leaf during plant growth. In general, a decrease was also detected, and, in the stem, the reduction was from 0.39% in Phase I to 0.30% in Phase IV. In leaf, the reduction in the content of β-D-glucan was mild, from an average content of 0.60% in Phase I to 0.54% in the last monitored phase. On the other hand, a massive increase in the content of β-D-glucan was observed in panicles with the plant’s maturation. From 0.52% in Phase II, the average content of β-D-glucan in panicles was raised to 2.22% in Phase IV.

The decrease suggests a function for β-D-glucan in cereals as a mobile energy source since β-D-glucan is composed of glucoside residues linked by β-1,3- and β-1,4-glycosidic linkages [26,39]. β-D-glucan fulfills this role as a temporary energy source [40] during the rapid growth of tissues due to the higher activity of β-D-glucan hydrolysis enzymes such as β-D-glucan endohydrolases observed in fast-growing tissues [41,42]. In barley, for example, the β-D-glucan in the walls could be used as a readily available source of metabolizable glucose during periods of darkness when glucanase rises glucose levels to hydrolyze the glucose [41]. The decrease in β-D-glucan content during plant development in selected vegetative tissues is probably also because the elasticity of the cell wall is no longer required, and hydrolysis occurs to recycle the carbon sources. This was the explanation for the scenario found in developing maize seedlings, where the β-D-glucan content was the highest when the cells were elongating and then declined when growth stopped [43].

Oppositely, an increase in the concentration of β-D-glucan has been studied during cell expansion, particularly in the elongating coleoptile. In the maize coleoptile, the concentration of β-D-glucan in the cell walls increased when the coleoptile was almost at the peak of elongation but decreased again after finishing the process [43]. A similar pattern of changes in β-D-glucan concentration in coleoptiles during development was also observed in barley [44]. In cereals, β-D-glucan is usually found in the expanding cells of organs such as the coleoptile [21,44], but may also be present in some vascular and fibrous cells of leaves [45], suggesting a structural role in secondary cell walls.

In our study, an increase in the content of β-D-glucan was detected in panicles during plant development. Panicles are the basis of grain development and in starch endosperm walls of the mature grain, which do not show secondary thickening, β-D-glucan may be present in high concentrations and may contribute up to 18% of the total glucose stored in the grain [46].

When comparing the β-D-glucan content in naked and hulled oats, the highest content of this polysaccharide was found in the naked oat varieties. The obtained results correlate with other works that have reached the same conclusions in the mature oat grains [47,48]. Although the conditions in which oats grow influence the β-D-glucan content of the tissue, the genotype has the most significant effect on the β-D-glucan content [34]. Higher content of β-D-glucan in naked grains, which are naturally not protected by the glume, could also be explained by the plant’s need to utilize β-D-glucan as a protective agent [49,50] or it could have been conditioned that caused stress to the plant (dehydration, pathogens, etc.) [51].

## 5. Conclusions

The β-D-glucan content was analyzed in four ontogenetic phases (BBCH 13, 30, 55, 71) during oat cultivation, in four plant organs (roots, stems, leaves and panicles) and in two hulled (Vaclav and Valentin) and two naked (Tatran and Hronec) oat varieties. The average content of the monitored biopolymer was 0.29 ± 0.14% in roots, 0.32 ± 0.11% in stems, 0.48 ± 0.13% in leaves and 1.28 ± 0.79% in panicles, respectively. In the root, a decrease in the content of β-D-glucan was observed from Phase I to Phase IV. At first, the highest content was observed in hulled oats (0.58%) compared to naked (0.44%) and finally the content was almost similar (0.19% and 0.17%, respectively). The highest content of β-D-glucan in the stem was in the BBCH 30 (Phase II) in hulled oats (0.48%) and in the first phase in naked oats (0.43%). In leaves during plant development, the content of β-D-glucan was comparable between Phases I and IV (on average 0.50% in both phases) with a decrease in Phases II (0.36% and 0.48% in hulled and naked oats, respectively) and III (0.34% and 0.38%) in all monitored oat varieties. The accumulation of β-D-glucan in panicles during oat ontogenesis showed an increasing trend, while in all oat varieties the content increased from 0.69% and 0.34% to 2.23% and 2.22% in both hulled and naked oat, respectively. The sampling date was the factor of variability in the content of β-D-glucan in root and panicle, in all monitored oat varieties. In the stem in hulled varieties and leaf in naked varieties, neither sampling date nor variety influenced the polysaccharide content. Understanding the behavior of the monitored biopolymer during plant development is necessary due to practical applications of the natural resources of β-D-glucan and research on the importance of the polysaccharide in the plant.

## Figures and Tables

**Figure 1 polymers-14-02668-f001:**
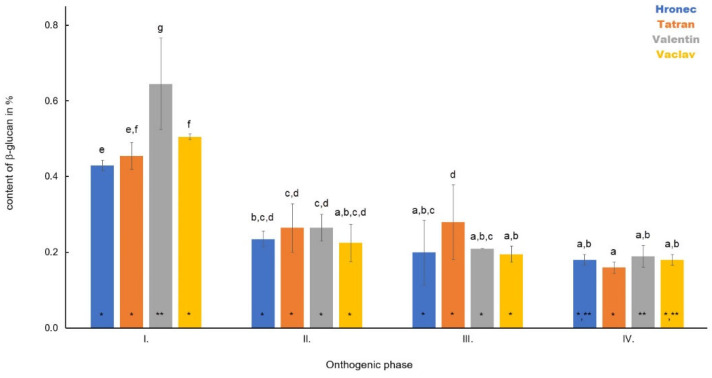
The dynamics in the content of β-D-glucan in roots of four analysed oat varieties during plant development. Different letters indicate a statistically significant difference between individual varieties in specific ontogenetic phase evaluated by ANOVA (LSD, *p* ≤ 0.05). Star symbols (* and **) represent statistically significant difference between four varieties evaluated for each monitored ontogenetic phase individually by ANOVA (LSD, *p* ≤ 0.05).

**Figure 2 polymers-14-02668-f002:**
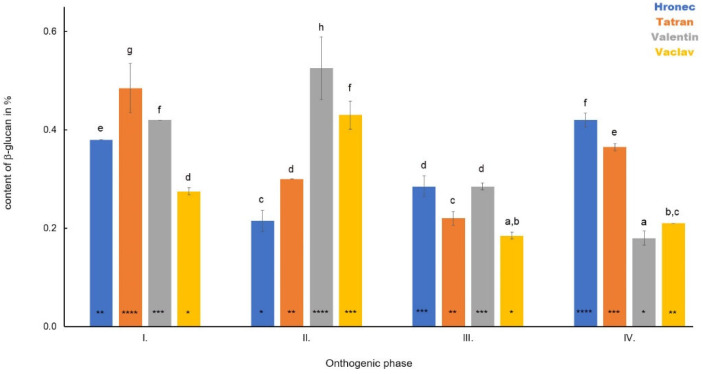
The dynamics in the content of β-D-glucan in stems of four analysed oat varieties during plant development. Different letters indicate a statistically significant difference between individual varieties in specific ontogenetic phase evaluated by ANOVA (LSD, *p* ≤ 0.05). Star symbols (*, **, ***, and ****) represent statistically significant difference between four varieties evaluated for each monitored ontogenetic phase individually by ANOVA (LSD, *p* ≤ 0.05).

**Figure 3 polymers-14-02668-f003:**
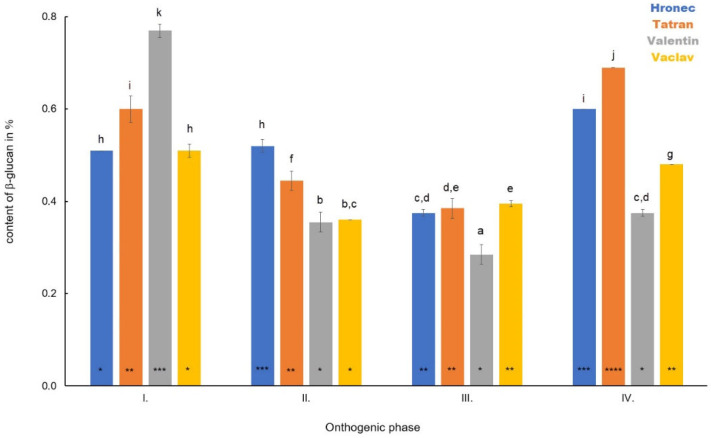
The dynamics in the content of β-D-glucan in leaves of four analysed oat varieties during plant development. Different letters indicate a statistically significant difference between individual varieties in specific ontogenetic phase evaluated by ANOVA (LSD, *p* ≤ 0.05). Star symbols (*, **, ***, and ****) represent statistically significant difference between four varieties evaluated for each monitored ontogenetic phase individually by ANOVA (LSD, *p* ≤ 0.05).

**Figure 4 polymers-14-02668-f004:**
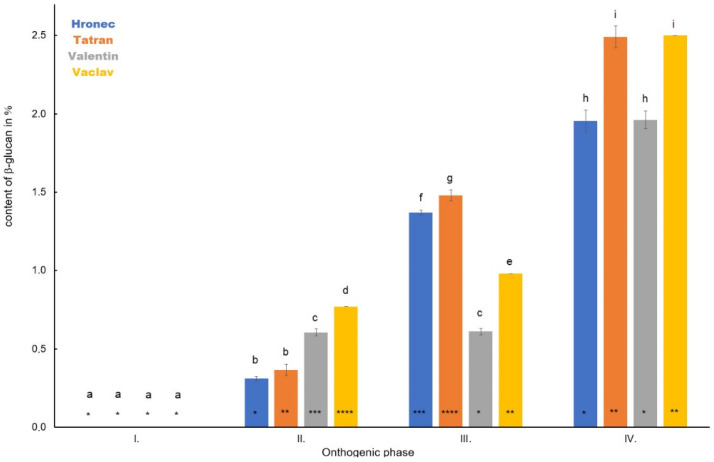
The dynamics in the content of β-D-glucan in panicles of four analysed oat varieties during plant development. Different letters indicate a statistically significant difference between individual varieties in specific ontogenetic phase evaluated by ANOVA (LSD, *p* ≤ 0.05). Star symbols (*, **, ***, and ****) represent statistically significant difference between four varieties evaluated for each monitored ontogenetic phase individually by ANOVA (LSD, *p* ≤ 0.05).

**Figure 5 polymers-14-02668-f005:**
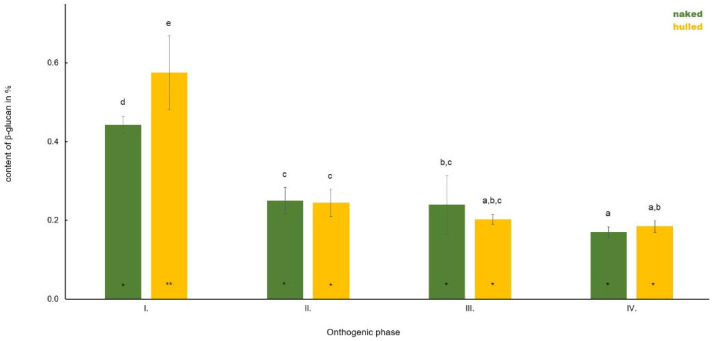
Comparison of the content of β-D-glucan in the root of hulled and naked oat samples during plant ontogenesis. Different letters indicate a statistically significant difference between individual seed types (naked and hull) in specific ontogenetic phase evaluated by ANOVA (LSD, *p* ≤ 0.05). Star symbols (* and **) represent statistically significant difference between naked and hulled samples evaluated for each ontogenetic phase individually by ANOVA (LSD, *p* ≤ 0.05).

**Figure 6 polymers-14-02668-f006:**
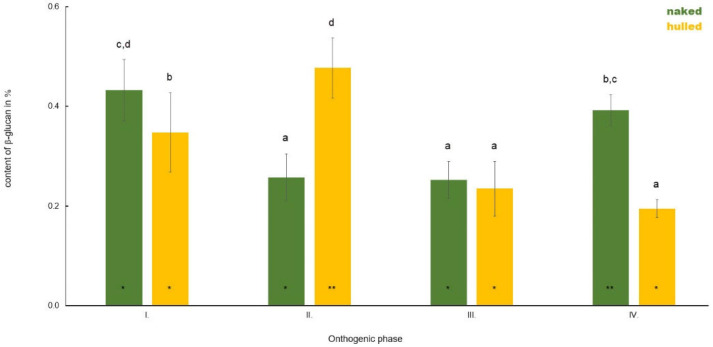
Comparison of the content of β-D-glucan in the stem of hulled and naked oat samples during plant ontogenesis. Different letters indicate a statistically significant difference between individual seed types (naked and hull) in specific ontogenetic phase evaluated by ANOVA (LSD, *p* ≤ 0.05). Star symbols (* and **) represent statistically significant difference between naked and hulled samples evaluated for each ontogenetic phase individually by ANOVA (LSD, *p* ≤ 0.05).

**Figure 7 polymers-14-02668-f007:**
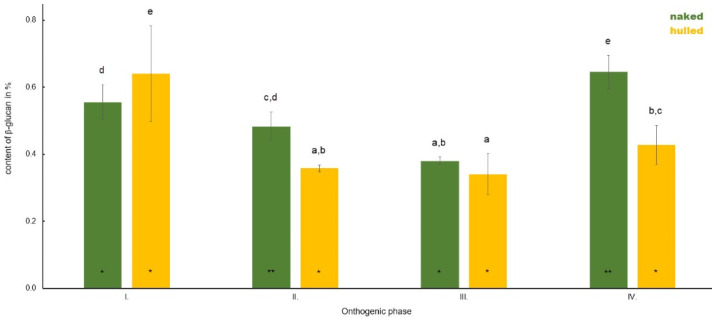
Comparison of the content of β-D-glucan in the leaf of hulled and naked oat samples during plant ontogenesis. Different letters indicate a statistically significant difference between individual seed types (naked and hull) in specific ontogenetic phase evaluated by ANOVA (LSD, *p* ≤ 0.05). Star symbols (* and **) represent statistically significant difference between naked and hulled samples evaluated for each ontogenetic phase individually by ANOVA (LSD, *p* ≤ 0.05).

**Figure 8 polymers-14-02668-f008:**
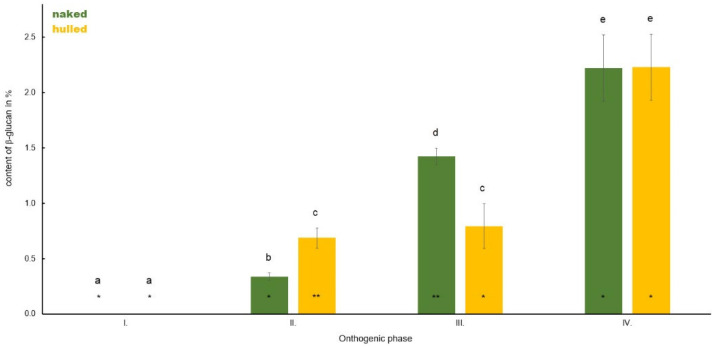
Comparison of the content of β-D-glucan in the panicle of hulled and naked oat samples during plant ontogenesis. Different letters indicate a statistically significant difference between individual seed types (naked and hull) in specific ontogenetic phase evaluated by ANOVA (LSD, *p* ≤ 0.05). Star symbols (* and **) represent statistically significant difference between naked and hulled samples evaluated for each ontogenetic phase individually by ANOVA (LSD, *p* ≤ 0.05).

## Data Availability

Not applicable.

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
