# Peer review of "Study of Dynamic Accumulation in β-D-Glucan in Oat (Avena sativa L.) during Plant Development"

_polymers, 2022, doi:10.3390/polym14132668_

Round 1

Reviewer 1 Report

The manuscript polymers-1764142 describes the quantification of B-D-glucan in organs collected from four oat varieties (hulled and naked ones) at different developmental stages. In my opinion, the manuscript in its present form is not suitable for publication in Polymers. My comments are reported directly in the manuscript as found in the attached file. 

Briefly, a few key points. 1) The English language needs thorough revising. 2) The (few) methods should be described more accurately. 3) Since the results are simply the output of one assay, it would have been preferred (and not much time consuming) if the data were ran in five, rather than three, replicates. 4) In the Results section, the comparisons between data on hulled and naked varieties are not properly handled (see my comments in the attached file). 5) The Introduction section is unnecessarily long and not well focused on the article and journal topic. 6) The Discussion is also not well focused on the results. 7) Some of the References I checked - and I did not check them thoroughly - are not proper. Too many references.

Author Response

Reviewer 1

The manuscript polymers-1764142 describes the quantification of B-D-glucan in organs collected from four oat varieties (hulled and naked ones) at different developmental stages. In my opinion, the manuscript in its present form is not suitable for publication in Polymers.

Answer: Thank you very much for your time and energy to carefully read the manuscript. Also, thank you very much for your useful comments to improve the quality of the manuscript.

My comments are reported directly in the manuscript as found in the attached file. 

Answer: Thank you for the comments. All of them were accepted and according to them the text was corrected.

Briefly, a few key points. 1) The English language needs thorough revising.

Answer: The English was carefully checked and according to both, your kind remarks and our effort and possibilities corrected.

2) The (few) methods should be described more accurately.

Answer: Methods mentioned in the text (sampling and glucan content analyses) were better and deeper described.

3) Since the results are simply the output of one assay, it would have been preferred (and not much time consuming) if the data were ran in five, rather than three, replicates.

Answer: We do agree that more replicates in experiment mean more reliability of presented results. On the other hand, we did not observe major differences in values variance, nor in repeated measurements, either between individual varieties tissues in each ontogenetic phase.  

4) In the Results section, the comparisons between data on hulled and naked varieties are not properly handled (see my comments in the attached file).

Answer: This part of the Results section was according to reviewer´s comments rewritten in the text. I hope, I understand your comment.

5) The Introduction section is unnecessarily long and not well focused on the article and journal topic.

Answer: The Introduction section was according to reviewer´s comments rewritten and according to other reviewers it was shorten. We tried to use as much relevant articles as we could find. There is not much research and not exactly focused on the accumulation of glucans during plant ontogenesis in vegetative organs.

6) The Discussion is also not well focused on the results.

Answer: The Discussion section was slightly rewritten to be better focussing on the results. Because there is not enough literature focusing exactly to our topis, in the Discussion section we did our best to corrected the text.   

7) Some of the References I checked - and I did not check them thoroughly - are not proper. Too many references.

Answer: The number of references was decreased; old references were removed and all of them were carefully corrected according to the instructions for authors.

Reviewer 2 Report

Reviewer report for paper entitled "Study of dynamic accumulation in β-D-glucan in oat (Avena sativa L.) during plant development" This paper was focused on in depth study the accumulation of one of the most important functional biopolymer (B-D-glucan) in oat. This research of special interest for both scientific and industrial societies. 

The research is well designed and the data presented of high quality and showed good reproducibility based on the statistical analysis shown in this work. However, some points I recommend to improve before publication of this work as follows:

- The introduction part need to be shorten by 20-25% and provide more focus on the previous studies directly related to this work (polysaccharides accumulation dynamic in plant cells). 

- In materials and methods part, need to give more details for the method of B-glucan determination/reducing sugar determination and provide all details of chemicals, and equipment used (Model, Brand, City, Country).

- The results part is well-written and data well presented in graph form. However, the font size in graph (both axes and numbers) need to be increase to be more readable in the final form. The experimental sequence is well prepared 

- Discussion part is acceptable. However, its recommended to do second round of improvement 

- Conclusion is missing, I recommend to provide strong conclusion at the end of this work. 

All in all, this work of high quality and provided new information about the dynamic of functional polysaccharides accumulation in plant and I recommend for acceptance after minor revision. 

Author Response

Reviewer 2

Reviewer report for paper entitled "Study of dynamic accumulation in β-D-glucan in oat (Avena sativa L.) during plant development" This paper was focused on in depth study the accumulation of one of the most important functional biopolymer (B-D-glucan) in oat. This research of special interest for both scientific and industrial societies. 

Answer: Thank you very much for the time and energy to read the manuscript and to improve its quality. Especially thank you for nice words about the quality and importance of the manuscript.

The research is well designed and the data presented of high quality and showed good reproducibility based on the statistical analysis shown in this work. However, some points I recommend to improve before publication of this work as follows:

- The introduction part need to be shorten by 20-25% and provide more focus on the previous studies directly related to this work (polysaccharides accumulation dynamic in plant cells). 

Answer: The Introduction part was shortened. Old references were removed, and some repeating and redundant sentences were deleted. Some old references we kept, because we think, they are important. To put new references better focused on the topis of the manuscript – ontogenesis and accumulation of glucans – were not possible to add, because there is really only few research published in this field.

- In materials and methods part, need to give more details for the method of B-glucan determination/reducing sugar determination and provide all details of chemicals, and equipment used (Model, Brand, City, Country).

Answer: The description of the method used to determine the content of glucans was rewritten to by more accurate, with more details. It was rewritten according to the procedure of Megazyme: https://www.megazyme.com/documents/Assay_Protocol/K-BGLU_DATA.pdf

- The results part is well-written and data well presented in graph form. However, the font size in graph (both axes and numbers) need to be increase to be more readable in the final form. The experimental sequence is well prepared 

Answer: The quality of figures was increased.

- Discussion part is acceptable. However, its recommended to do second round of improvement 

Answer: The Discussion section was carefully checked and according to other two reviewers corrected to be improved.

- Conclusion is missing, I recommend to provide strong conclusion at the end of this work. 

Answer: The Conclusion section was added to the manuscript.

All in all, this work of high quality and provided new information about the dynamic of functional polysaccharides accumulation in plant and I recommend for acceptance after minor revision. 

Answer: Thank you very much for this nice and positive evaluation. Especially for my PhD. student Veronika Gregusová it this evaluation very motivational for the next work.

Reviewer 3 Report

Abstract part, please include more data information and statistical analysis results.

The authors should describe in a more concrete way the method of determination of beta-glucans. It is indicated that an aliquot is taken, how much?.  A phosphate buffer is used, at what concentration?.

Has a standard line been made for the determination of glucose by the oxidase-peroxidase method?

How the authors calculate the % of beta-glucans? Has the moisture content of the samples been taken into account?

Why have both ANOVA and GLM been used as statistical analysis?

Improve image quality.

There are too many old references. It is recommended to use references from the last 5 years.

The conclusions of the work should be improved.

Author Response

Reviewer 3

Abstract part, please include more data information and statistical analysis results.

Answer: The Abstract section was corrected, and more results were included.

The authors should describe in a more concrete way the method of determination of beta-glucans. It is indicated that an aliquot is taken, how much?.  A phosphate buffer is used, at what concentration?.

Answer: The method used in the work to analyze the content of beta-D-glucans was described more deeper and concrete. The text was added according the megazyme procedure: https://www.megazyme.com/documents/Assay_Protocol/K-BGLU_DATA.pdf

Has a standard line been made for the determination of glucose by the oxidase-peroxidase method?

Answer: Yes, according to the methodology and the procedure, a standard line was used for the determination of glucose. If the reviewer would be so kind, here is the process described and we followed it exactly step by step. https://www.megazyme.com/documents/Assay_Protocol/K-BGLU_DATA.pdf

How the authors calculate the % of beta-glucans? Has the moisture content of the samples been taken into account?

Answer: The content of beta-D-glucans was calculated according to the formula below described in the assay kit document. The moisture content was measured in few samples. Because the samples were carefully dried to 100% dry weight and analyzed immediately after drying, we considered the dry matter to be 100% and thus did not recalculate the resulting glucan content.

Why have both ANOVA and GLM been used as statistical analysis?

Answer: We are sorry for this unfortunate mistake. Results presented in the manuscript are all from ANOVA with post hoc test (LSD), although the GLM was used in preliminary data evaluation, the results were not satisfactory. 

Improve image quality.

Answer: The quality of figures was improved. I hope, the quality of pictures is OK now. If not, I am very sorry, in that case, please write me and I will correct them better. 

There are too many old references. It is recommended to use references from the last 5 years.

Answer: The number of references was decreased and some of old and inappropriate references were removed. However, we have retained some older references as we see them appropriate. We have not been able to find more recent research on these references.

The conclusions of the work should be improved.

Answer: The Conclusion section was added to the manuscript.

Round 2

Reviewer 1 Report

Comments are to be found in the attached file

Author Response

Dear reviewer, I would like to express my strong thanks to your time and energy to read (not only once) the manuscript and to give us so many good remarks to improve the quality of the text. I really appreciate it. 

In the final version I marked all changes and all of your comments and suggestions were aplied in the final version. Once again, many thanks for your time. 

Reviewer 3 Report

All proposed corrections have been made.

Author Response

Dear reviewer, I would like to express my strong thanks to your time and energy to read caferully (not only one time) the manuscript. I really appreciate it. I hope, the final version of the manuscript is of good quality. 

Round 3

Reviewer 1 Report

The Authors have accepted and executed all of the suggestion made; except for some minor errors in the English language - which I suggest to check once more before publication -, I find the manuscript suitable for publication in Polymers